# CRISPR-Cas13a-Based Assay for Accurate Detection of OXA-48 and GES Carbapenemases

Concha Ortiz-Cartagena,[a,c] Daniel Pablo-Marcos,[b] Laura Fernández-García,[a,c] Lucía Blasco,[a,c] Olga Pacios,[a,c] Inés Bleriot,[a,c] María Siller,[b] María López,[a,c] Javier Fernández,[d,i] Belén Aracil,[e,j] Pablo Arturo Fraile-Ribot,[f,j] Sergio García-Fernández,[b] Felipe Fernández-Cuenca,[h] Marta Hernández-García,[g,j] Rafael Cantón,[g,j] Jorge Calvo-Montes,[b,j] María Tomás[a,c]

[a]Multidisciplinary and Translational Microbiology Group (MicroTM), Biomedical Research Institute of A Coruña (INIBIC), Microbiology Service, University Hospital of A Coruña (CHUAC), University of A Coruña (UDC), A Coruña, Spain

[b]Microbiology Service, University Hospital Marqués de Valdecilla – IDIVAL, Santander, Spain

[c]Study Group on Mechanisms of Action and Resistance to Antimicrobials (GEMARA) on behalf of the Spanish Society of Infectious Diseases and Clinical Microbiology (SEIMC), Madrid, Spain

[d]Microbiology Service, University Hospital Central de Asturias. Translational Microbiology Group, ISPA, Oviedo, Spain

[e]Reference and Research Laboratory for Antibiotic Resistance and Health Care Infections, National Centre for Microbiology, Institute of Health Carlos III, Majadahonda, Madrid, Spain

[f]Microbiology Service, University Hospital Son Espases and Health Research Institute Illes Balears (IdISBa), Palma de Mallorca, Spain

[g]Microbiology Service, University Hospital Ramón y Cajal and Ramón y Cajal Health Research Institute (IRYCIS), Madrid, Spain

[h]Clinical Unit of Infectious Diseases and Microbiology, University Hospital Virgen Macarena, Institute of Biomedicine of Sevilla (University Hospital Virgen Macarena/CSIC/University of Sevilla), Sevilla, Spain

[i]CIBER de Enfermedades Respiratorias (CIBERES), Instituto de Salud Carlos III, Madrid, Spain

[j]CIBER de Enfermedades Infecciosas (CIBERINFEC), Instituto de Salud Carlos III, Madrid, Spain

Concha Ortiz-Cartagena and Daniel Pablo-Marcos contributed equally to the work. The order was determined by the corresponding author after negotiation.

**ABSTRACT** Carbapenem-resistant pathogens have been recognized as a health concern as they are both difficult to treat and detect in clinical microbiology laboratories. Researchers are making great efforts to develop highly specific, sensitive, accurate, and rapid diagnostic techniques, required to prevent the spread of these microorganisms and improve the prognosis of patients. In this context, CRISPR-Cas systems are proposed as promising tools for the development of diagnostic methods due to their high specificity; the Cas13a endonuclease can discriminate single nucleotide changes and displays collateral cleavage activity against single-stranded RNA molecules when activated. This technology is usually combined with isothermal pre-amplification reactions in order to increase its sensitivity. We have developed a new LAMP-CRISPR-Cas13a-based assay for the detection of OXA-48 and GES carbapenemases in clinical samples without the need for nucleic acid purification and concentration. To evaluate the assay, we used 68 OXA-48-like-producing *Klebsiella pneumoniae* clinical isolates as well as 64 *Enterobacter cloacae* complex GES-6, 14 *Pseudomonas aeruginosa* GES-5, 9 *Serratia marcescens* GES-6, 5 *P. aeruginosa* GES-6, and 3 *P. aeruginosa* (GES-15, GES-27, and GES-40) and 1 *K. pneumoniae* GES-2 isolates. The assay, which takes less than 2 h and costs approximately 10 € per reaction, exhibited 100% specificity and sensitivity (99% confidence interval [CI]) for both OXA-48 and all GES carbapenemases.

**IMPORTANCE** Carbapenems are one of the last-resort antibiotics for defense against multidrug-resistant pathogens. Multiple nucleic acid amplification methods, including multiplex PCR, multiplex loop-mediated isothermal amplification (LAMP) and multiplex RPAs, can achieve rapid, accurate, and simultaneous detection of several resistance genes to carbapenems in a single reaction. However, these assays need thermal cycling steps and specialized instruments, giving them limited application in the field. In this work, we adapted with high specificity and sensitivity values, a new LAMP CRISPR-Cas13a-based

Address correspondence to María Tomás, MA.del.Mar.Tomas.Carmona@sergas.es.

The authors declare no conflict of interest.

assay for the detection of OXA-48 and GES carbapenemases in clinical samples without the need for RNA extraction.

**KEYWORDS** detection, GES, OXA-48, LAMP, CRISPR-Cas13a

Bacterial antimicrobial resistance has emerged as a public health threat in the 21st century (1, 2), and it is estimated that bacterial resistance will cause 10 million deaths by 2050 (3). Furthermore, the involvement of resistance pathogens can lead to long-term hospitalization of patients, often requiring expensive and intensive care. The ESKAPE pathogens (*Enterococcus faecium*, *Staphylococcus aureus*, *Klebsiella pneumoniae*, *Acinetobacter baumannii*, *Pseudomonas aeruginosa*, and *Enterobacter* spp.) are particularly important due to their enhanced ability to develop antimicrobial resistance through plasmid acquisition (1). Carbapenems have become drugs of last resort for treating severe infections caused by Gram-negative bacteria (4, 5). Worryingly, reports of carbapenemase-producing organisms (CPO), which are included in the list of priority pathogens compiled by the World Health Organization (WHO), are constantly increasing (3). Accurate early detection is therefore crucial to prevent the spread of CPOs and improve the prognosis of patients (2, 6, 7).

Some class A (KPC) and class B (VIM/IMP/NDM) carbapenemases usually confer high levels of resistance to carbapenems and most ß-lactams (8). The powerful hydrolytic activity of these enzymes enables them to be easily detected in clinical microbiology laboratories by phenotypic screening methods, such as the carbapenem inhibition method (CIM) (9). However, class D carbapenemases (OXA-48) and the so-called "minor class A carbapenemases" (Guiana extended-spectrum [GES]) exhibit lower levels of carbapenem hydrolysis, which increases the false negative rate of the phenotypic methods, complicating detection of these groups (6) and requiring a confirmatory molecular test (2, 5). These validating tests are usually based on qPCR amplification (e.g., Amplidiag CarbaR+MCR Kit, EntericBio CPE, BD Max System), but also, on Loop-Mediated Isothermal Amplification (LAMP) (e.g., Eazyplex SuperBug CRE), Luminex (e.g., Luminex xTAG) and multiplex microarray technologies, something that raises costs and delays the reporting of laboratory results. Thus, there is an urgent need to develop a highly specific, sensitive, rapid and simple method to detect these carbapenemases.

Isothermal amplification reactions combined with clustered regularly interspaced short palindromic repeats (CRISPR) and their associated proteins (Cas) are becoming essential to detect multidrug resistant microorganisms (10, 11) as they improve the specificity of diagnostic tests. Although different methods of isothermal amplification are available, LAMP displays greater sensitivity and specificity than other methods (12, 13) and has previously been used to detect several pathogens (12, 14, 15).

Among the wide variety of Cas endonucleases, Cas13a (C2c2) nuclease provides several advantages regarding the development of diagnostic techniques. On the one hand, Cas13a is highly specific and is able to discriminate between two sequences varying in only a single nucleotide (16). On the other hand, the fact that it acts on single-stranded RNA (ssRNA) molecules could be used to detect active infections. Moreover, once Cas13a is specifically activated by RNA target binding, its so-called collateral cleavage activity will indiscriminately cleave any ssRNA molecules (10, 11). These features indicate the CRISPR-Cas13a system as a promising tool for the development of diagnostic tests.

In a recent study, our research group developed a LAMP-CRISPR-Cas13a technique free from nucleic acid purification and concentration for SARS-CoV-2 detection in nasopharyngeal samples. The results, revealed in HybriDetect lateral flow strips, yielded 83% sensitivity and 100% specificity (17). In the present study, we applied the same technology to clinical samples from different sources, leading to the successful detection of both carbapenem-resistance $bla_{OXA-48}$ and $bla_{GES}$ genes (Fig. 1). Moreover, an optimistic comparative study was carried out with the main molecular techniques currently available to detect the $bla_{OXA-48}$ and $bla_{GES}$ genes, in terms of: sensitivity, specificity, extraction sophistication, time, and cost per reaction.

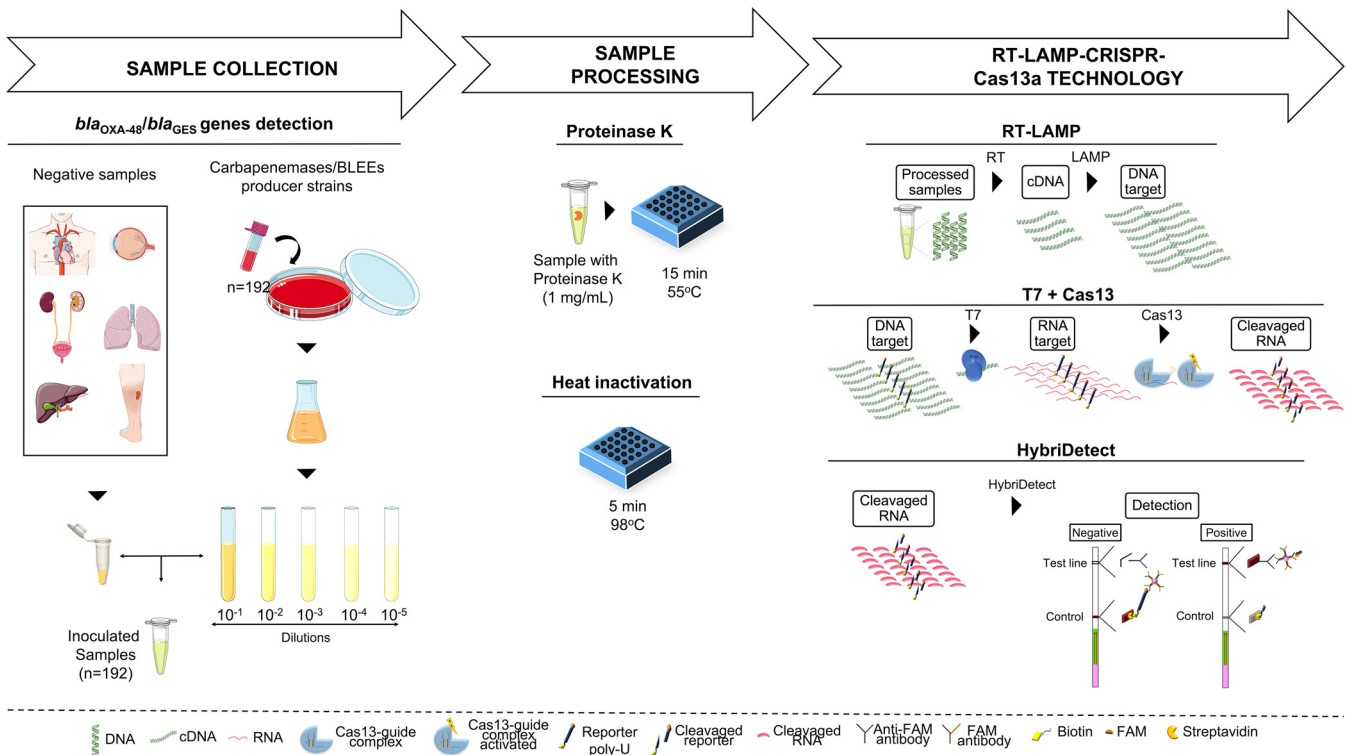

**FIG 1** Workflow of the protocol for diagnosis of infectious diseases based on the CRISPR-Cas13a system.

## RESULTS

**Study of the state of the art.** A total of 12 studies on molecular diagnostics of different carbapenemases were found; the $bla_{OXA-48}$ gene was detected in all of these studies, and the $bla_{GES}$ gene was detected in seven of them (18–24). Regarding the diagnostic technologies on which these techniques were based, seven out of 12 articles were based on multiplex qPCR (Check Direct CPE, Amplidiag CarbaR+MCR Kit, EntericBio CPE, BD Max System and in-house reactions) (20–26), two out of 12 protocols were based on LAMP reaction (Eazyplex SuperBug CRE and LAMP-HNB) (27, 28), and three out of 12 were based on Luminex system (Luminex xTAG) (19), multiplex conventional PCR (in-house reaction) (18) and multiplex microarray technology, respectively (29).

The comparative study revealed that, in this context of molecular diagnostic of carbapenem-resistant bacteria, our LAMP-CRISPR-Cas13a technology is a strong rival to the current diagnostic kit that are available in terms of quality-price ratio.

**Limit of detection.** The limit of detection (LoD) values obtained for each clinical isolate group are shown in Fig. 2.

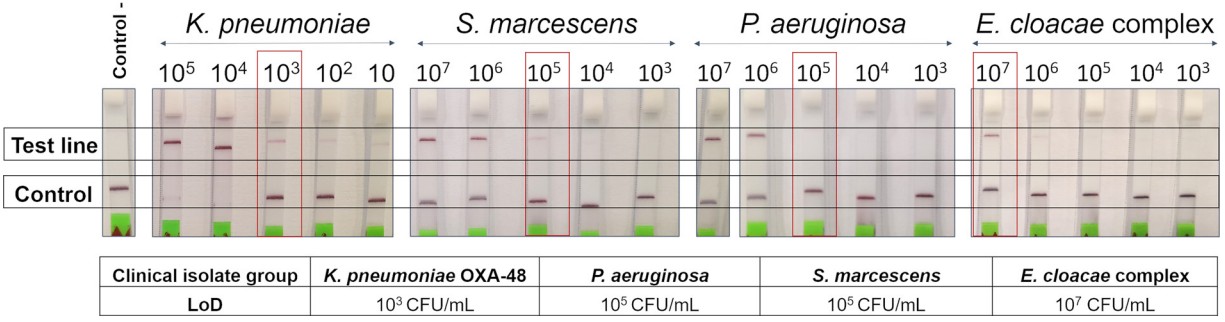

| Clinical isolate group | K. pneumoniae OXA-48 | P. aeruginosa | S. marcescens | E. cloacae complex |
|---|---|---|---|---|
| LoD | $10^3$ CFU/mL | $10^5$ CFU/mL | $10^5$ CFU/mL | $10^7$ CFU/mL |

**FIG 2** LoD assay for OXA-48 and GES-producing bacteria inoculated in clinical negative samples that the LAMP-CRISPR-Cas13a technique detects in serial dilutions (1:10) of LB cultures. Thin gray toolings indicate strips spliced for labeling purposes.

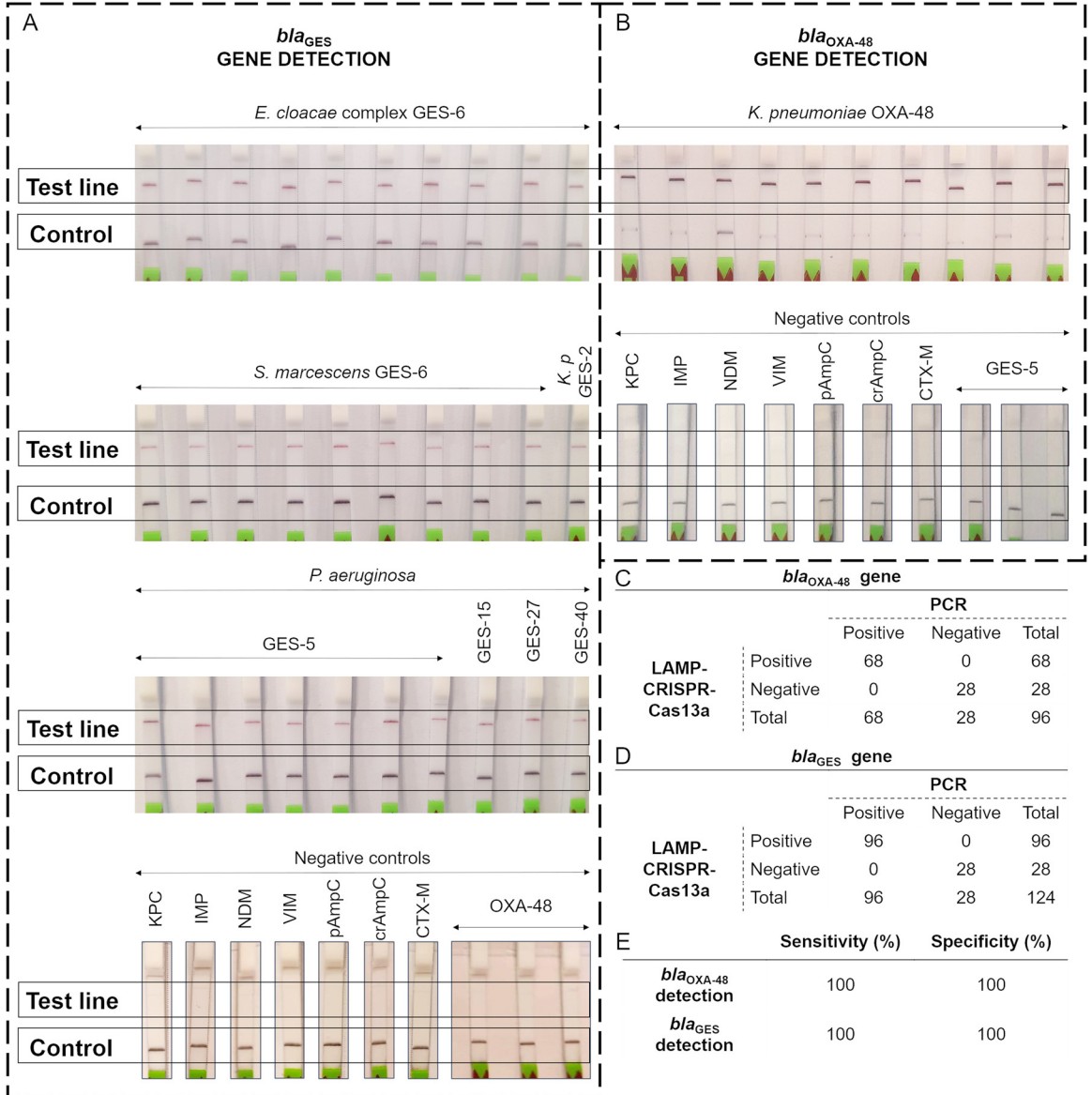

**FIG 3** (A) HybriDetect lateral flow test strip for GES carbapenemase identification by $bla_{GES}$ gene detection in inoculated samples and non GES carbapenemase-producing strains (negative controls). (B) HybriDetect lateral flow test strip for OXA-48 carbapenemase identification by $bla_{OXA-48}$ gene detection in inoculated samples and non OXA-48 carbapenemase-producing strains (negative controls). (C) Results obtained for $bla_{GES}$ gene detection. (D) Results obtained for $bla_{OXA-48}$ gene detection. (E) Table including the specificity and sensitivity (99% CI) for the LAMP-CRISPR-Cas13a technique calculated by processing data in Fig. 3C and 3D. Thin gray toolings indicate strips spliced for labeling purposes.

**Detection of OXA-48 and GES carbapenemases.** This novel LAMP-CRISPR-Cas13a assay, which takes less than 2 h and costs approximately 10 € per reaction, was able to detect the $bla_{OXA-48}$ gene in all *K. pneumoniae* OXA-48 tested in the inoculated samples. Similarly, the $bla_{GES}$ gene was also detected in all the samples contaminated with the strains harboring GES carbapenemase.

Furthermore, none of other carbapenemases tested (VIM, IMP, KPC, and NDM, used as negative controls) were detected in the negative controls. Indeed, no other antibiotic resistance mechanisms (ESBL CTX-M-group, plasmidic AmpC, or hyperproduction of chromosomal AmpC) produced cross-reactivity.

Based on the results obtained (Fig. 3), we estimated that our LAMP-CRISPR-Cas13a-based method exhibits 100% specificity and sensitivity (99% confidence interval [CI]) for detection of both OXA-48 and all GES carbapenemases.

## DISCUSSION

State of the art study revealed that greater efforts must be made to innovate in genotypic diagnostic methods for carbapenem resistance, due to the important impact on public health and to the current lack of a rapid sensitive, specific technique (Table S1). This is especially worrying in the case of GES carbapenemases, as most commercial kits for carbapenemase detection do not include this enzyme class, also reflected by the scarce number of papers concerning this topic. Indeed, some kits, such as Amplidiag CarbaR+MCR kit, failed to detect GES-carbapenemase-producing organisms. In addition, as already mentioned, despite the presence of GES or OXA-48 carbapenemases, the carbapenem MIC values may be within the range of susceptibility due to the high variability in expression (6).

Our research group has developed a LAMP-CRISPR-Cas13a-based protocol that allows easy detection of this kind of carbapenemases in both the Enterobacterales and *Pseudomonas* spp. in clinical microbiology laboratories. This promising assay fulfils the three key features of accuracy, accessibility, and affordability considered by WHO (30). Thus, on the one hand, it showed an accuracy of 100% for detection of both $bla_{OXA-48}$ and all $bla_{GES}$ genes. Furthermore, it is both accessible and affordable as neither specific equipment nor trained personnel are required, and the amounts of enzymes needed per reaction are quite low. In fact, in terms of cost per reaction, this CRISPR-Cas based assay is highly competitive with most commercial kits, as shown in Table S1; only comparable with the LAMP-HNB technique, which shows some disadvantages in comparison to this LAMP-CRISPR-Cas13a protocol, as said below.

This study revealed a wide range of LoD values, varying from $10^3$ CFU/mL of *K. pneumoniae* OXA-48 to $10^7$ CFU/mL of *E. cloacae* complex GES-6. This may be due to differences in the number of copies of the plasmids harboring $bla_{OXA-48}$ and $bla_{GES}$ genes. Recent studies have reported a lower LoD of $10^2$ CFU/mL in *K. pneumoniae* OXA-48 (26), although this can be explained by the fact that the CRISPR-Cas systems add a level of specificity to the diagnostic tests that the LAMP reaction alone does not have. On the contrary, other studies reported a higher LoD, also in *K. pneumoniae* OXA-48, in the order of $10^4$ CFU/mL by LAMP reaction or $10^9$ CFU/mL by qPCR (28).

The comparative study revealed that the Eazyplex SuperBug CRE system (27) produced the best results, with 100% values of sensitivity and specificity in only 15 min. In addition, five out of seven multiplex qPCR assays (21–24, 26) yielded excellent results, with parameter values of 100%, although they were slower (1 to 3 h). Likewise, very promising results were achieved with Luminex technology and even multiplex conventional PCR (specificity of 99.4% to 100% and sensitivity of 100%), although the assays were also slower (2 to 5 h) (18, 19). By contrast, poorer results were obtained with the other two multiplex qPCR assays (Amplidiag CarbaR+MCR and Check-Direct CPE) (20, 25), which yielded sensitivity of 92% to 100% and specificity of 88 to 100, and took up to 3 h. Finally, multiplex microarray-based technology yielded a sensitivity of 98.3%, specificity of 99.6%; and was faster than multiplex qPCR (2 h) (29).

Comparison of the sensitivity and specificity of our assay with those obtained with other assays showed that our results are at least as good as those of other LAMP assays and most multiplex qPCR assays (18, 21–24, 26–28). In addition, we obtained higher specificity values than the Check-Direct CPE (25) and Luminex xTAG assay (19), as well as higher sensitivity values than the Amplidiag CarbaR+MCR kit (20) (Table S1). Furthermore, our technique yielded higher sensitivity and specificity values than in the multiplex microarray study reviewed (29).

Three of the aforementioned assays that yielded the best results (Eazyplex SuperBug CRE, BD Max System, and LAMP-HNB) used fluorescent and colorimetric dyes, respectively, which bind non-specifically to double-stranded DNA (22, 27, 28), without corroboration of the sequence-specificity of the amplicon. This is worrisome due to the possible low specificity of the amplification. Moreover, some fluorescent dyes (e.g., SYBR green) inhibit the LAMP reaction and therefore must be added at the end of the assay, thus increasing the risk of contamination (28). Hopefully, CRISPR-Cas assays use specific probes that increase the specificity of the diagnostic tests.

In summary, the high levels of specificity and sensitivity obtained with this promising nucleic acid purification and concentration-free protocol make the LAMP-CRISPR-Cas13a-

**TABLE 1** Sequences of primers, crRNAs, and reporters

| Name | | |
|---|---|---|
| **LAMP primers**[a] | **Sequence 5'->3'** | **Gene position** |
| F3_GES | GATCGTCGAATGGTCTCCTG | 285 to 304 |
| B3_GES | GCCATAGCAATAGGCGTAGT | 526 to 545 |
| FIP_GES | GCTAAGCTGCACCGCAGCTT<u>GAAATTAATACGACTCACTATAGGG</u>GTTTCTAGCATCGGGACACA | - |
| BIP_GES | AGAAATTGGCGGACCTGCTGCGGTCTAGCCGACTCACAGA | - |
| Floop_GES | CGGGCTCCAGCTTCCAGAGATA | 335 to 353 |
| Bloop_GES | CGCAGTATTTTCGTAAAATTGGCG | 431 to 454 |
| F3_OXA-48 | TCTTAAACGGGCGAACCA | 174 to 191 |
| B3_OXA-48 | ATACGTGCCTCGCCAATT | 387 to 404 |
| FIP_OXA-48 | GCGTCTGTCCATCCCACTTAAAGAC<u>GAAATTAATACGACTCACTATAGGG</u>AATAGCTTGATCGCCCTCG | - |
| BIP_OXA-48 | CGCCACTTGGAATCGCGATCATGGCGGGCAAATTCTTGA | - |
| Floop_OXA-48 | GCTGCCTCGAGAACCGTCA | 246 to 269 |
| Bloop_OXA-48 | CCGCGATGAAATATTCAGTTGTGCC | 338 to 362 |
| **crRNA** | | |
| crRNA_GES | gauuuagacuacccaaaaacgaaggggacuaaaacCUCAGUAAGAGGUUAGUAGCCCCAUUGU | 377-404 |
| crRNA_OXA-48 | gauuuagacuacccaaaaacgaaggggacuaaaacGCAAAUUCUUGAUAAACAGGCACAACUG | 352-380 |
| **Reporters** | | |
| Reporter | FAM-UUUUUU-Biotin | - |

[a]Underlined letters indicate the T7 promoter sequence and lower-case letters indicate the scaffold sequence. All primers were supplied by IDT, and reporters were supplied by GenScript.

based assay one of the best rapid and specific diagnostic methods for antibiotic resistance genes and infectious diseases. The technique could be established as a diagnostic tool for routine testing in clinical microbiology laboratories to detect other multi-drug resistant pathogens. Concerning the current limitations, Cas13 detection methods should be optimized to avoid prior amplification of nucleic acids. This future perspective would allow a 70% reduction in the cost per reaction and the direct identification of target RNA fragments to confirm the expression of these genes, besides the original detection protocol described above.

## MATERIALS AND METHODS

**Study of the state of the art.** A state of the art study was performed to compare the use of different classical diagnostic methods for OXA-48 and GES carbapenemases and the novel LAMP-CRISPR-Cas13a-based assay in terms of specificity, sensitivity, protocol extraction, time, and costs. We compiled several studies of genotypic diagnostic technologies for detection of the $bla_{OXA-48}$ and $bla_{GES}$ genes. The specificity and sensitivity were calculated to evaluate the effectiveness of the LAMP-CRISPR-Cas13a-assay compared to existing assays and economic data were obtained from the UK Health Security Agency to compare affordability.

**Nucleic acid preparations.** The reference gene sequences for OXA-48 and GES beta-lactamases were assembled from GenBank (https://www.ncbi.nlm.nih.gov/genbank). The target sequences were analyzed *in silico* and specific primers were designed in order to amplify a conserved genetic region (174 to 404 pb $bla_{OXA-48}$ gene region and 285 to 544 pb $bla_{GES}$ gene region). Thus, three pairs of LAMP primers were designed using PrimerExplorer V5 software (F3/B3, FIP/BIP, Floop/Bloop). The FIP LAMP primers included the T7 polymerase promoter in their sequences for the subsequent transcription step. Finally, crRNA molecules were designed to hybridize with the highly conserved regions of the LAMP amplicons of both genes (Table 1).

The RNA reporter molecules, designed for signal amplification, harbor a single isomer derivative of fluorescein modification (FAM) at the 5' extreme and a biotin molecule at the 3' extreme.

**Clinical samples.** Numerous isolates were used to assess the effectiveness of the CRISPR-Cas13a assay. Thus, a total of 68 OXA-48-like-producing *K. pneumoniae* clinical isolates and 96 GES-producing Gram-negative bacteria, both previously characterized by PCR, were collected for the assay. Among the last, 64 *E. cloacae* complex GES-6, 14 *P. aeruginosa* GES-5, 9 *S. marcescens* GES-6, 5 *P. aeruginosa* GES-6, and 3 *P. aeruginosa* (GES-15, GES-27, and GES-40) and 1 *K. pneumoniae* GES-2 (Table 2).

In addition, 28 clinical isolates with diverse resistance mechanisms were used as negative controls: 4 KPC, 4 NDM, 4 IMP, 4 VIM, 4 ESBL CTX-M-group, 4 plasmidic AmpC, and 4 hyperproducers of the chromosomal AmpC (Table 2).

Several negative clinical samples (urine, bronchial aspirate, bronchial secretion, bronchial biopsy, abscess, biliary fluid, venous ulcer, pleural fluid) were obtained with the purpose of contaminating them with each strain.

**Contamination with carbapenemase-producing strains.** The negative clinical samples were contaminated at random with one dilution up of the limit of detection of each clinical isolate group to produce a more reliable signal and better interpretation of the results (see *Limit of detection* section below). Each negative sample was inoculated separately with more than one clinical strain. To this end, clinical isolates were grown in

**TABLE 2** Clinical isolates used for evaluation of the CRISPR-Cas13a assay

| OXA-48/GES molecular detection | Microorganism | Resistance mechanism |
|---|---|---|
| Positive | *Klebsiella pneumoniae* (N = 68) | OXA-48 |
| | *K. pneumoniae* (N = 1) | GES-2 |
| | *P. aeruginosa* (N = 14) | GES-5 |
| | *Enterobacter cloacae* complex (N = 64) | GES-6 |
| | *Serratia marcescens* (N = 9) | |
| | *Pseudomonas aeruginosa* (N = 5) | |
| | *P. aeruginosa* (N = 1) | GES-15 |
| | *P. aeruginosa* (N = 1) | GES-27 |
| | *P. aeruginosa* (N = 1) | GES-40 |
| Negative | *K. pneumoniae* (N = 2) | KPC |
| | *Citrobacter freundii* complex (N = 2) | |
| | *E. cloacae* complex (N = 3) | IMP |
| | *P. aeruginosa* (N = 1) | |
| | *Escherichia coli* (N = 2) | NDM |
| | *K. pneumoniae* (N = 1) | |
| | *C. freundii* complex (N = 1) | |
| | *Pseudomonas putida* (N = 3) | VIM |
| | *E. cloacae* complex (N = 1) | |
| | *E. coli* (N = 2) | pAmpC |
| | *Proteus mirabilis* (N = 1) | |
| | *K. pneumoniae* (N = 1) | |
| | *E. coli* (N = 3) | ESBL CTX-M-group |
| | *K. pneumoniae* (N = 1) | |
| | *E. coli* (N = 1) | Hyperproduction of the chromosomal AmpC |
| | *C. freundii* complex (N = 1) | |
| | *Morganella morganii* (N = 1) | |
| | *Klebsiella aerogenes* (N = 1) | |

Luria-Bertani medium (LB) (0.5% NaCl, 0.5% yeast extract, 1% tryptone) until an optical density (OD) corresponding to $10^8$ CFU/mL was reached. The isolates were then serially diluted in saline solution to yield a concentration of one logarithm higher than what was intended to be inoculated. Finally, 20 $\mu$L of each dilution was mixed with 180 $\mu$L of a negative sample, which is an additional 1:10 dilution.

**Sample processing.** Nucleic acid extraction was performed following the proteinase K-heat inactivation protocol (PK-HID) (31). According to the instructions, 5 $\mu$L of proteinase K (10 mg/mL, Thermo Fisher Scientific, Waltham, MA, USA) was added to 95 $\mu$L of each sample. Finally, the mixed sample was placed in a constant-temperature metal water bath at 55°C for 15 min, followed by 5 min of heat inactivation at 98°C.

**LAMP reaction.** LAMP DNA and RNA amplification (WarmStart LAMP Kit [DNA & RNA], NEB, Ipswich, MA, USA) was performed following the manufacturer's protocol. Briefly, processed samples (5 $\mu$L) were added to a reaction mixture containing 12.5 $\mu$L of WarmStart LAMP 2x Master Mix and 2.5 $\mu$L of Primer Mix 10x (FIP/BIP 16 $\mu$M, F3/B3 2 $\mu$M, LOOPF/LOOPB 4 $\mu$M, stock) adjusted to a final volume of 25 $\mu$L with RNase-free water. The samples with reaction mixture were then incubated at 65°C for 1 h.

**Collateral cleavage activity-based detection.** Each Cas13a-based detection reaction was carried out at 37°C for 30 min with the following reaction components: 2 $\mu$L of cleavage buffer 10X (GenCRISPR Cas13a (C2c2) Nuclease, GenScript), 0.5 $\mu$L of dNTPs (HiScribe T7 Quick High Yield RNA Synthesis Kit, NEB), 0.5 $\mu$L of T7 polymerase (HiScribe T7 Quick High Yield RNA Synthesis Kit, NEB), 20 U of RNase murine inhibitor (NEB), 0.015 $\mu$L of *Leptotrichia wadei*, LwaCas13a endonuclease (25 nM, GenCRISPR Cas13a [C2c2] Nuclease, GenScript), 0.5 $\mu$L of crRNA (50 nM, IDT), 2 $\mu$L of reporter (1,000 nM, GenScript), and 5 $\mu$L of cDNA sample, adjusted to a final volume of 20 $\mu$L with RNase-free water.

**Hybridetect lateral flow.** The test results were revealed by the HybriDetect lateral flow test strip method, as described by the manufacturer (Milenia Biotec, Giessen, Germany), with some modifications. Briefly, 20 $\mu$L of collateral cleavage activity-based detection product was mixed with 80 $\mu$L of assay buffer supplemented with 5% polyethylenglycol in a 96-well plate where strip tests were placed. The test results are visible to the naked eye after 2 to 3 min.

**Limit of detection.** In order to determine the LoD, one strain of each clinical isolate group was used to generate reference growth curves (1 *K. pneumoniae* OXA-48, 1 *E. cloacae* complex GES-6, 1 *P. aeruginosa* GES-5, and 1 *S. marcescens* GES-6). Thus, after overnight culture, a 1:100 dilution in LB was used to initiate a batch culture that allowed us to generate a standard curve for each clinical isolate group, relating CFU and OD. The OD gives an instant approximation of bacterial cell count measured in a spectrophotometer. The batch culture was measured at several OD values (0.1, 0.2, 0.3, 0.6, and 0.9) and was serially diluted 1:10 in saline solution to yield $10^8$ to $10^2$ CFU/mL of each OD measured. Aliquots (100 $\mu$L) from each dilution tube were spread plated and incubated. The dilution plate that produced 10 to 200 colonies was used to calculate the CFU/mL for the given OD. Finally, the LoD of each strain was calculated using the CRISPR-Cas technique in contaminated, negative clinical samples at each concentration ($10^8$ to $10^2$ CFU/mL).

## SUPPLEMENTAL MATERIAL

Supplemental material is available online only.

**SUPPLEMENTAL FILE 1**, PDF file, 0.05 MB.

## ACKNOWLEDGMENTS

This study was funded by grants PI19/00878 and PI22/00323 awarded to M. Tomás, within the State Plan for R+D+I 2013-2016 (National Plan for Scientific Research, Technological Development and Innovation 2008-2011) and co-financed by the ISCIII-Deputy General Directorate for Evaluation and Promotion of Research - European Regional Development Fund "A Way of Making Europe" and Instituto de Salud Carlos III FEDER, Spanish Network for the Research in Infectious Diseases (REIPI, RD16/0016/0006 and CIBER CB21/13/00012, CB21/13/00084 and CB21/13/00095), Instituto de Salud Carlos III FEDER. The research was also funded by grant IN607D 2021/10 within the GAIN (Agencia Gallega de Innovación) and by the Study Group on Mechanisms of Action and Resistance to Antimicrobials, GEMARA (SEIMC, http://www.seimc.org/), and finally an ESCMID grant (European Society of Clinical Microbiology and Infectious Diseases) awarded to L. Fernández-García. O. Pacios, L. Fernández-García, and M. López were financially supported by the grants IN606A-2020/035, IN606B-2021/013 and IN606C-2022/002, respectively (GAIN, Xunta de Galicia). D. Pablo-Marcos was financially supported by a López Albo grant. I. Bleriot was financially supported by pFIS pro- 511 g (ISCIII, FI20/00302).

We declare that there are no conflicts of interest.

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
