## [Reviewer comments · Microbiology Spectrum]

Microbiology Spectrum

CRISPR-Cas13a-based assay for accurate detection of OXA-48 and GES carbapenemases

Concha Ortiz-Cartagena, Daniel Pablo-Marcos, Laura Fernandez-Garcia, Lucia Blasco, Olga Pacios, Inés Bleriot, Maria Siller, María López Díaz, Javier Fernández, Belen Aracil, Pablo Fraile-Ribot, Sergio Garcia-Fernández, Felipe Fernández-Cuenca, Marta Hernández-García, Rafael Cantón, Jorge Calvo-Montes, and Maria Tomas

Corresponding Author(s): Maria Tomas, INIBIC-CHUAC

Review Timeline:

Submission Date:	March 28, 2023
Editorial Decision:	June 6, 2023
Revision Received:	June 16, 2023
Accepted:	June 22, 2023

Editor: Rebecca Shapiro

Reviewer(s): Disclosure of reviewer identity is with reference to reviewer comments included in decision letter(s). The following individuals involved in review of your submission have agreed to reveal their identity: Rodolfo Garcia-Contreras (Reviewer #2)

Transaction Report:

DOI: <https://doi.org/10.1128/spectrum.01329-23>

June 6, 2023

Dr. Maria Tomas
INIBIC-CHUAC
Microbiology
As Xubias 84
A Coruña
Spain

Re: Spectrum01329-23 (CRISPR-Cas13a-based assay for accurate detection of OXA-48 and GES carbapenemases)

Dear Dr. Maria Tomas:

Thank you for submitting your paper to Microbiology Spectrum. Two reviewers have now read your manuscript and have made a few important suggestions that will need to be addressed before we consider the manuscript for possible publication.

Link Not Available

Sincerely,

Rebecca Shapiro

Journals Department
Reviewer comments:

Reviewer #1 (Comments for the Author):

The manuscript describes the performance of a new technique for rapid and accurate detection of OXA-48 and GES carbapenemases that could make a positive impact on diagnosis of carbapenem-resistant pathogens and patient prognosis. The manuscript is of interest since it addresses the important topic of diagnosis improvement to combat the global major health threat of antibiotic resistance.

However, there are some major and minor issues that the authors should address before manuscript publication.

Major issues:

Abstract and Manuscript. The authors state that the technique circumvents DNA extraction. They should modify this statement since RNA extraction does take place by Proteinase K-heat inactivation. A more precise statement would be to indicate that the technique does not require purification and concentration step, thus allowing simpler and faster workflow.

Abstract and Manuscript. The limit of detection value and the values of sensitivity and specificity for each carbapenem-resistant gene target should be informed separately. Values of sensitivity and specificity should include 95% confidence interval lower and upper values. Positive and negative predictive values should not be informed, since authors used a convenience collection of strains isolated from clinical samples obtained from different sources and not a collection of strains from clinical samples consecutively isolated.

Manuscript, Results section. State-of-the-art techniques should be explained succinctly in the Introduction section.

Manuscript. M&M section. Comparative performance of state-of-the-art techniques should be discussed in the Discussion section and Table 1 should be placed in a Supplemental File, since it does not refer to results of performance of the technique that is the subject of study.

Minor issues

-Tables 2 and 3 should be reordered as Table 1 and 2, respectively.

Reviewer #2 (Comments for the Author):

I think the work presented is interesting and the new methodology very promising for the easy and rapid detection of carbapenemase genes, I just have minor concerns:

- 1) please add in the discussion a comparison of the cost of your method compared with others.
- 2) all the methods can detect the carbapenemase genes, but still the bacteria may not express them, for example if they contain nonsense, missense mutations, is this true? if so, please add this to the discussion.
- 3) L 38 Clinical Microbiology (no need of capital letters).
- 4) L 57 Multiple, rather than multiples.

Staff Comments:

Preparing Revision Guidelines

For complete guidelines on revision requirements, please see the journal Submission and Review Process requirements at <https://journals.asm.org/journal/Spectrum/submission-review-process>. **Submissions of a paper that does not conform to**

Microbiology Spectrum guidelines will delay acceptance of your manuscript. "

Please return the manuscript within 60 days; if you cannot complete the modification within this time period, please contact me. If you do not wish to modify the manuscript and prefer to submit it to another journal, please notify me of your decision immediately so that the manuscript may be formally withdrawn from consideration by Microbiology Spectrum.

Reviewer comments

Reviewer #1 (Comments for the Author):

The manuscript describes the performance of a new technique for rapid and accurate detection of OXA-48 and GES carbapenemases that could make a positive impact on diagnosis of carbapenem-resistant pathogens and patient prognosis. The manuscript is of interest since it addresses the important topic of diagnosis improvement to combat the global major health threat of antibiotic resistance.

However, there are some major and minor issues that the authors should address before manuscript publication.

Major issues:

1) Abstract and Manuscript. The authors state that the technique circumvents DNA extraction. They should modify this statement since RNA extraction does take place by Proteinase K-heat inactivation. A more precise statement would be to indicate that the technique does not require purification and concentration step, thus allowing simpler and faster workflow.

-Thank you for the comment, corrections were made in lines: 49, 117 and 222; and in Supplementary Table S1.

2) Abstract and Manuscript. The limit of detection value and the values of sensitivity and specificity for each carbapenem-resistant gene target should be informed separately. Values of sensitivity and specificity should include 95% confidence interval lower and upper values. Positive and negative predictive values should not be informed, since authors used a convenience collection of strains isolated from clinical samples obtained from different sources and not a collection of strains from clinical samples consecutively isolated.

-Thank you for this comment.

On the one hand, we designed a unic crRNA, which is common to all the *bla*_{GES} genes, because we think that this idea presents more clinical relevance than the detection of every *bla*_{GES} genes separately. That is why we calculated the limit of detection of every bacterial specie, since we only pretended to know the minimum concentration (CFU/mL) of bacterial inoculated to be detected. We have decided to modify figure 2 for not to deal to confusion.

On the other hand, 99 % confidence interval has been added in lines 55, 156-157 and 480; and PPV and NPV have been eliminated from the document.

3) Manuscript, Results section. State-of-the-art techniques should be explained succinctly in the Introduction section.

-Thank you for this comment, we really think it improves the understanding of the article. Corrections were made in lines: 92-96 and 122-125.

4) Manuscript. M&M section. Comparative performance of state-of-the-art techniques should be discussed in the Discussion section and Table 1 should be placed in a Supplemental File, since it does not refer to results of performance of the technique that is the subject of study.

-Thank you for this comment, we really think it improves the article. Corrections were made in lines: 130-141, 162, 180, 192-203, 209, 494 and 495.

Minor issues:

Tables 2 and 3 should be reordered as Table 1 and 2, respectively.

- Corrections were made in lines: 252, 263, 266, 490 and 493.

Reviewer #2 (Comments for the Author):

I think the work presented is interesting and the new methodology very promising for the easy and rapid detection of carbapenemase genes, I just have **minor concerns**:

1) please add in the discussion a comparison of the cost of your method compared with others.

-Thank you for the comment, it really adds an interesting point of view. Corrections were made in lines: 54, 125, 146-147, 178-182, 237, 240-241 and 497-500; and in supplementary table S1.

2) all the methods can detect the carbapenemase genes, but still the bacteria may not express them, for example if they contain nonsense, missense mutations, is this true? if so, please add this to the discussion.

-Thank you for the comment, correction was made in lines: 227-231.

3) L 38 Clinical Microbiology (no need of capital letters).

-Corrections were made in lines: 38, 86, 172 and 225-226.

4) L 57 Multiple, rather than multiples.

-Thank you, correction was made in line 59

June 22, 2023

Dr. Maria Tomas
INIBIC-CHUAC
Microbiology
As Xubias 84
A Coruña
Spain

Re: Spectrum01329-23R1 (CRISPR-Cas13a-based assay for accurate detection of OXA-48 and GES carbapenemases)

Dear Dr. Maria Tomas:

Thank you for your revised submission to Microbiology Spectrum, we are pleased to inform you that your manuscript has now been accepted. I am forwarding it to the ASM Journals Department for publication. You will be notified when your proofs are ready to be viewed.

Sincerely,

Rebecca Shapiro
Editor, Microbiology Spectrum
